# Blood-Derived Liquid Biopsies Using Foundation One^®^ Liquid CDx for Children and Adolescents with High-Risk Malignancies: A Monocentric Experience

**DOI:** 10.3390/cancers14112774

**Published:** 2022-06-02

**Authors:** Fanny Cahn, Gabriel Revon-Riviere, Victoria Min, Angélique Rome, Pauline Filaine, Annick Pelletier, Sylvie Abed, Jean-Claude Gentet, Arnauld Verschuur, Nicolas André

**Affiliations:** 1Department of Pediatric Oncology, La Timone University Hospital of Marseille, Assistance Publique-Hôpitaux de Marseille, 13005 Marseille, France; fanny.cahn@ap-hm.fr (F.C.); gabriel.revon-riviere@sickkids.ca (G.R.-R.); victoria.min@ap-hm.fr (V.M.); angelique.rome@ap-hm.fr (A.R.); pauline.filaine@ap-hm.fr (P.F.); sylvie.abed@ap-hm.fr (S.A.); jean-claude.gentet@ap-hm.fr (J.-C.G.); arnauld.verschuur@ap-hm.fr (A.V.); 2Centre d’essais Précoces en Cancérologie de Marseille (CEPCM), CLIPP2, Assistance Publique-Hôpitaux de Marseille, 13005 Marseille, France; annick.pelletier@ap-hm.fr; 3SMARTc Unit, CRCM Inserm 1068, CNRS UMR 7258, Aix-Marseille University, 13005 Marseille, France

**Keywords:** precision medicine, liquid biopsy, pediatric oncology, targeted therapy, immunotherapies

## Abstract

**Simple Summary:**

Precision oncology requires tumor molecular profiling to identify actionable targets. Blood-derived liquid biopsy (LB) is a potential alternative that is not yet documented in real-world settings, especially in pediatric oncology. Analyzing, retrospectively, the use of LB in children with refractory relapsing diseases, we were able to show that this is a feasible alternative to tissue biopsy, resulting in successful analysis in a subset of patients.

**Abstract:**

Precision oncology requires tumor molecular profiling to identify actionable targets. Tumor biopsies are considered as the gold standard, but their indications are limited by the burden of procedures in children. Blood-derived liquid biopsy (LB) is a potential alternative that is not yet documented in real-world settings, especially in pediatric oncology. We performed a retrospective analysis of children and teenagers with a relapsing or refractory disease, upon whom LB was performed using the Foundation One® liquid CDx from 1 January 2020 to 31 December 2021 in a single center. Forty-five patients (27 boys) were included, with a median age of 9 years of age (range: 1.5–17 years old). Underlying malignancies were neuroblastoma (12 patients), bone sarcoma (12), soft tissue sarcoma (9), brain tumors (7), and miscellaneous tumors (5). Forty-three patients had metastatic disease. Six patients had more than one biopsy because of a failure in first LB. Median time to obtain results was 13 days. Overall, analysis was successful for 33/45 patients. Eight patients did not present any molecular abnormalities. Molecular alterations were identified in 25 samples with a mean of 2.1 alterations per sample. The most common alterations concerned TP53 (7 pts), EWS-FLI1 (5), ALK (3), MYC (3), and CREBBP (2). TMB was low in all cases. Six patients received treatment based on the results from LB analysis and all were treated off-trial. Three additional patients were included in early phase clinical trials. Mean duration of treatment was 85 days, with one patient with stable disease after eight months. Molecular profiling using Foundation One® Liquid CDx was feasible in pediatric patients with high-risk solid tumors and lead to identification of targetable mutations in a subset of patients.

## 1. Introduction

Cancer remains the main cause of disease-related mortality in children and adolescents in high-income countries [1]. It is now recognized that new approaches are needed to improve survival rates and reduce the burden of late effects of cancer therapy. Molecular profiling of tumors can lead to the identification of molecular targets that, in turn, can be actioned by new agents [2,3,4,5,6]. The concept of “cancer precision medicine” is now being implemented to guide treatment in patients with advanced malignancy in pediatric malignancies though several programs worldwide [7]. Precision medicine can lead to impactful modification of treatment, as shown recently by the INFORM registry and the MAPPYACTS trial [8,9]. In these recent reports, patients with the highest target priority level benefited from matched targeted treatment. The INFORM registry reported a median progression-free survival of 204 days, as compared to 117 days for all other patients, while the MAPPYACTS trial reported a 38% response rate in this subset of patients. Elsewhere, the Ped MATCH programs [10] enrolls children in molecularly adapted early phase trials.

Precision oncology requires biopsies of tumors to identify actionable targets exposing vulnerable patients to additional anesthesia, ionizing radiation, pain, and increasing admission time. Biopsies have been reported to have a complication rate in pediatrics of 6% to 8% [9,11]. The biopsy of primary tumors in advanced cancer patients may be declined or not offered in order to minimize the burden of care for these vulnerable patients, leading to a decreased proportion of profiled tumors. Circulating cell-free DNA (cfDNA)-analysis, also referred as liquid biopsy (LB), can be used in several body fluids without the limitations and inconvenience of tissue biopsies. Experience and knowledge in LB are increasing rapidly in the adult setting, with evidence of its clinical impact [12,13,14]. In children, feasibility of LB has been described in research settings with successful molecular profiling [15,16,17,18,19,20]. However, data is still quite limited, as well as reports on the real-world feasibility and usefulness of this tool. In the meantime, the FDA has approved Foundation One® Liquid CDx (F1LCDx) [21], paving the way for its use in daily practice in pediatric patients with high-risk solid tumors using a commercially available test.

We report our retrospective experience evaluating the feasibility, profiling results, and clinical impact of blood derived LB for children and adolescents with high-risk malignancies using F1LCDx.

## 2. Patients and Methods

### 2.1. Patients

We included all patients who underwent blood-derived LB through peripheral blood samples using F1LCDx from 1 January 2020 to 31 December 2021 in La Timone Children Hospital, AP-HM Marseille. This profiling was available at our center for patients with a relapsed or refractory solid extra-cerebral or cerebral tumor.

### 2.2. Genomic Analyses

Genomic analyses were performed using F1LCDx test, an NGS-based in vitro diagnostic tool analyzing a panel of 324 genes, using circulating cell-free DNA isolated from plasma-derived anti-coagulated peripheral whole blood. Blood samples were collected in two tubes of whole blood (8.5 mL per tube). Samples were then shipped at ambient temperature to Foundation Medicine, Inc. (Cambridge, MA, USA). The F1LCDx assay employs a single DNA extraction method to obtain cfDNA from plasma from whole blood. Genomic-signature-like tumor mutational burden (TMB), microsatellite instability (MSI) and tumor fraction were also reported [21].

A test was considered negative when it reported a sample failure or a lack of sufficient quantity of DNA. We defined as positive a test that was technically successful and for which a molecular profiling could be established. Actionable molecular alterations (AMA) were defined as molecular alterations for which a therapy targeting either the MA or the pathway activated by the MA was available, using the OncoKB database (v3.4; 17 June 2021) [22]. Tumor Mutational Burden (TMB) was considered “TMB-high” when the score was ≥10 mutations/Mb [9]. The decision to treat a patient was made via national or local molecular tumor board. We did not limit the report to clearly-defined oncogenic driver events with straightforward treatment recommendations, but also sought to describe alterations in genes that could lead to a potential benefit according to preclinical findings or clinical data in adult cancers, as per E-Smart Protocol [9].

Genomic analyses on solid tumors were performed via national profiling programs (MAPPYACTS or Michado). Details of the techniques used are described elsewhere [9].

### 2.3. Data

Clinical data were obtained from electronic clinical charts from patients and included age, gender, pathology, date of first diagnosis, date of last relapse, presence of metastasis, previous lines of treatment, known genomic alterations from a previous profiling method, and current and subsequent anti-cancer treatment based on the results from the LB.

## 3. Results

### 3.1. Patients

Forty-five patients (27 males/18 females) were included with a median age of 11 years (range 1.5–18). Characteristics of the study population, pathology, successful profiling, and molecular findings are summarized in Table 1.

### 3.2. Feasibility

The flow chart of the samples is depicted in Figure 1. Overall, out of the 51 samples evaluated in 45 patients, 33 lead to successful analysis. Six patients had more than one sample because of test failures. We performed another liquid biopsy for six of these patients and all lead to another failure. Only one LB performed among seven patients with brain tumors was successful. Importantly, results were available within a median time of 13 days (range 8–19).

### 3.3. Molecular Alterations and Genes Alterations Found with LB

Molecular alterations were identified in 25 samples. In eight patients, no molecular finding was reported. As described in Table 1 and Figure 2, among the alterations found in 25 LB, the most common alterations were TP53 mutations (7 pts), EWS1 fusions (5) with FL1 (4) or WT1 (1), and CREBBP (2). TMB was low in all cases (range 0–8) (Figure 1). A median number of two alterations was found with a maximum of five. The TP53 gene alterations were found in different types of tumors: neuroblastoma (1), Ewing sarcoma (1), rhabdomyosarcoma (1), osteosarcoma (1), medulloblastoma (1), melanoma (1), and adenocarcinoma (1). EWSR1 was found in 4/5 Ewing’s sarcomas. In neuroblastoma patients, profiling could be obtained in 10 out of 12 patients.

### 3.4. Molecular Alterations and Genes Alterations Found with Tissue Biopsies

Data from patients who underwent paired tissue and liquid tumor biopsies were collected. Nine patients were identified. The data have been included in Figure 2. Furthermore, as shown with a Venn diagram (Figure 3), a 31% overlap with 11 alterations in common between tumor and plasma was found. Besides, 10 molecular alterations were found on LB only and 14 on tissue biopsies only.

### 3.5. Clinical Impact for Patients

Six patients received treatment based on the results from LB analysis. Two patients with neuroblastoma harboring, respectively, an NF1 mutation and BRAF mutation were treated with trametinib. Trastuzumab combined with chemotherapy was given to one patient with neuroblastoma with ERBB2 amplification. One patient with a metastatic melanoma was treated with a combination of antiPD1–antiCTLA4. We did not have the TMB value for this patient, but the decision was made based on adult data on melanoma. One patient with Ewing sarcoma received a combination of olaparib and irinotecan as per molecular enrichment criteria of the E-Smart protocol arm D. None of these five patients were included in a clinical trial; they all received off-label therapy. One last patient with osteosarcoma was treated within a phase 1 trial following the discovery of a RAD amplification. All patients progressed quickly after one cycle of treatment, except for one patient with neuroblastoma who displayed a BRAF G469A mutation and was treated with trametinib on compassionate access. One patient experienced a sustained stable disease, leading to treatment for 9 months. Two additional patients with Ewing sarcoma and neuroblastoma satisfied inclusion criteria for a clinical trial of immunotherapy (Arm J Lirilumab-nivolumab Esmart) after being profiled by LB (non-enriched cohort). Both progressed after, respectively, 1 and 4 cycles of treatment. Overall, the mean duration of treatment was 85 days, with one patient with stable disease after 9 months.

## 4. Discussion

We report a real-world experience of high-throughput molecular profiling based on blood-derived LB in children and adolescents with high-risk malignancies using the commercially available F1LCDx test. To the best of our knowledge, this is the first report of the clinical use of this tool in pediatric oncology. We obtained molecular profiling of the tumor in 75% of the cases.

The high proportion of successful profiling in high-risk patients is likely in part to be explained by the number of metastatic patients in our population. This is consistent with previous findings where ctDNA levels were correlated with tumor burden [9,16]. In our population, neuroblastoma patients had the highest proportion of success for LB profiling, which is also consistent with previous studies [16]. Thus, this strategy allowed tissue biopsy at relapse for only a few patients with neuroblastoma.

Another interesting point is the turnaround time from biopsy to report. The FMI LB test required 13 days in our case, versus 25 days for Van Tilburg et al. [8] and 26 days for Harttrampf et al. [11], but in a single-center setting. Overall, this preliminary report suggests that LB is well-suited for disseminated pediatric malignancies and, particularly, relapsed or refractory neuroblastomas, to obtain molecular information on the tumor to make therapeutic decisions in a clinically relevant timeframe.

We found that blood-derived liquid biopsies did not frequently lead to successful analysis in patients with brain tumors. Moreover, in two patients for whom the analysis was technically successful, certain molecular characteristics of the tumors, such as the H3K27 mutation or P53 mutations found on the analysis of the tumoral tissue, were not found on LB. This is consistent with the results by Pages et al., who reported the very low efficiency of plasma-derived LB to identify mutations for patients with brain tumors [20]. Similarly, Izquierdo et al. reported that CSF-derived LB was more reliable than blood-derived LB in patients with high-grade and diffuse midline glioma [19]. It is therefore very likely that for brain tumors, cerebrospinal-fluid-derived LB will be further developed as a biomarker of residual disease, as recently reported in medulloblastoma [23].

In our study, 45 children had liquid biopsy sampled through peripheral blood, for whom 33 (74%) of which analysis could be technically performed. Mutations were identified in 57% of the cases and targetable alterations were found for 38% of the patients. Ultimately, 13% of the patients were treated according to the results of the molecular profiling. Large profiling programs relying on tissue biopsies have been reported and describe the spectrum of mutations and TMB in pediatric solid tumors [24]. Although our results cannot be formally compared, and although we mostly included non-cerebral and metastatic solid tumors, most frequent molecular alterations identified are consistent with our findings with P53 as the most frequently found alteration. George et al. reported targetable alterations ranging from 61% to 46% [25], with heterogeneous criteria to define actionable molecular alteration targets, which is consistent with our finding using LB. TMB was consistently low, as previously reported in large series of pediatric patients [9]. Of note, there are no data comparing tissue-based vs. liquid biopsy-based TMB.

There is currently very little pediatric data reporting the use of liquid biopsy for a wide panel of genetic alterations in pediatric oncology. Stankunaite et al. reported a series of 39 children with solid tumors, for whom they were able to perform analysis of 67 genes [17]. Interestingly, molecular abnormalities were found in all patients with extra-cranial malignancies and not under treatment using LB. They also reported consistent findings between tissue biopsies and LB. Similarly, Van Paemel et al. recently reported concordant results between LB or tissue biopsies when analyzing copy number alteration in a wide panel of pediatric solid tumors [18]. Elsewhere, Berlanga et al. have very recently published the results of the European MAPPYACTS trial [9]. Among the 787 patients who were included, more than one genetic alteration leading to targeted treatment suggestion was identified in the 436 patients (69%) for whom successful sequencing could be achieved. Interestingly, cell-free DNA extracted from the plasma of 234 patients with extracerebral tumors led to the detection in 76% of actionable alterations also found in tumor tissue, further strengthening LB for molecular profiling as a reliable alternative to tumor tissue biopsy. In adults, recently, analysis of cell-free DNA extracted from the blood is a recognized tool in non-small-cell lung cancer international practice guidelines [26], mostly for disease monitoring, and provides several advantages compared to analysis performed via tissue biopsy [27]. A recent study by Gouton et al. was the first to evaluate the feasibility and clinical impact of liquid biopsy for cell-free DNA-based NGS analysis in pan-cancer patients, also using F1LCDx [28].

In our experience, among the nine patients who had both LB and tissue biopsies, there were 31% common alterations; 40% of alterations found on tissue biopsies only and 29% of alterations found on LB only. Although these results are based a small subset of patients among the series we report here, and although LB and tumor sampling was not conducted during the course of the disease, this might illustrate the absence of pediatric enhancement of the Foundation One^®^ Liquid CDx test. As previously described, some genomic events enriched in pediatric cancer may not be captured by adult panels [17]. This represents a risk of underestimating and missing an opportunity to treat an actionable target, although according to OncoKB classification, Class 1, 2, and 3 alterations were missed using the LB. Further research is warranted to address this gap.

This study was also limited by its retrospective and monocentric design, leading to well-identified biases and limitations. The study population shows a high proportion of patients with metastatic solid tumors patients, a high proportion of neuroblastoma, and only few CNS tumors. However, as no restriction on any characteristics were imposed by inclusion criteria, this study describes the use of LB in a real-world setting.

LB may contribute to filling the gap to profile the tumors of patients with high-risk malignancies for whom tumor biopsy at relapse may be difficult or refused by parents/patients. Although we did not report any measurable response or clear improvement in progression-free survival in our sample, we highlighted that 25 out of 45 high-risk patients did not access tissue-based biopsy at relapse. Out of these 25, 14 patients received a successful profiling using LB, and five saw their treatment plan changed. The potential benefit of finding high-priority targets is highlighted by one patient who was successfully treated with a MEK inhibitor [29]. We therefore advocate for LB to be performed in the context of high-risk malignancy and further explored, as it may represent a genuine alternative to tissue biopsy. Additionally, as suggested by Stankunaite et al. and previous authors, Ct DNA in LB could also be used to monitor response to treatment [16,17,23].

To demonstrate that targeting specific mutations or pathways can lead to lead to meaningful clinical benefit in pediatric oncology beyond “ready for use” alterations is a critical issue. Thus, the INFORM registry (8) has recently demonstrated that in patients harboring the highest target priority level (42 patients—8.1%), the 20 patients who received matched targeted therapy reached a median progression-free survival of 204 days, compared to 117 days (*p* = 0.011) in all other patients. Similarly, the arms E and F of the E-smart trial reported that eight out of nine patients with stable disease displayed alterations considered for enrichment in 25 patients upon treatment of ribociclib in combination with temozolomide–topotecan or everolimus. [30]. Elsewhere, the first pediatric MATCH treatment arm to be completed has reported limited efficacy of selumetinib in a cohort of patients with treatment-refractory tumors harboring MAPK alterations [31]. Of note, MEK inhibitors have previously demonstrated promising responses, for instance, in low-grade glioma and plexiform neurofibroma. Similarly, the E and F arms of the Esmart trial failed to demonstrate any response after treatment with vistusertib alone or in combination with temozolomide and topotecan [32]. Altogether, these results suggest that pathway mutation status alone may not be insufficient to predict response to a given monotherapy/combination for refractory/relapsing pediatric cancers.

## 5. Conclusions

Overall, we believe that LB will contribute to a breakthrough change in clinical practice to achieve precision medicine for children and adolescents with cancer. High-throughput molecular profiling is feasible in clinical routine in a reasonable timeframe with molecular findings consistent with tissue-based analyses. This could impact care for high-risk pediatric patients by avoiding unnecessary burden and shortening the time for results in difficult clinical situations. Prospective and state-of-the-art evaluation in larger cohorts is mandatory.

## Figures and Tables

**Figure 1 cancers-14-02774-f001:**
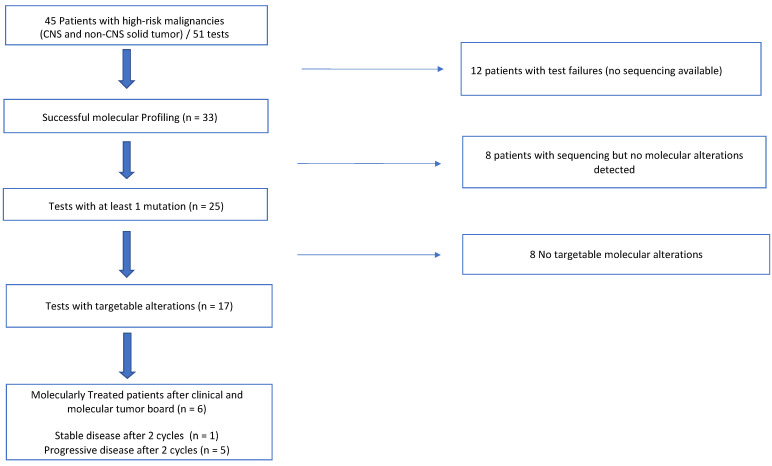
Flowchart of patients who underwent liquid biopsies.

**Figure 2 cancers-14-02774-f002:**
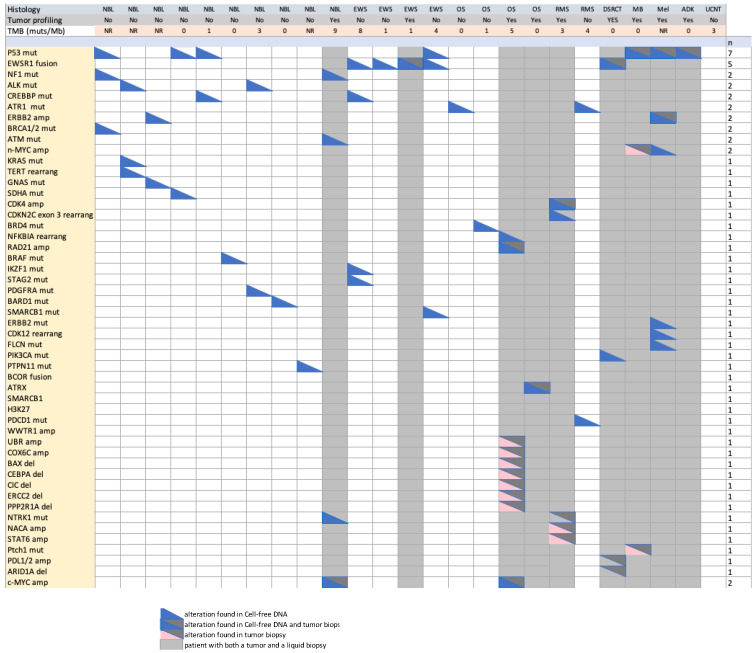
Oncoplot of molecular alteration and TMN found in liquid biopsies and tumor biopsies. Abbreviations used: TMB, tumor mutational burden; amp, amplification; rearrang, rearrangement; NR, not reported in LB results. NBL: neuroblastoma; EWS: Ewing sarcoma, OS: osteosarcoma; RMS: rhabdomyosarcoma; MB: medulloblastoma; DSCRT: desmoplastic small round cell tumor; Mel: melanoma; ADK: adenocarcinoma; UCNT: undifferentiated carcinoma of the nasopharynx.

**Figure 3 cancers-14-02774-f003:**
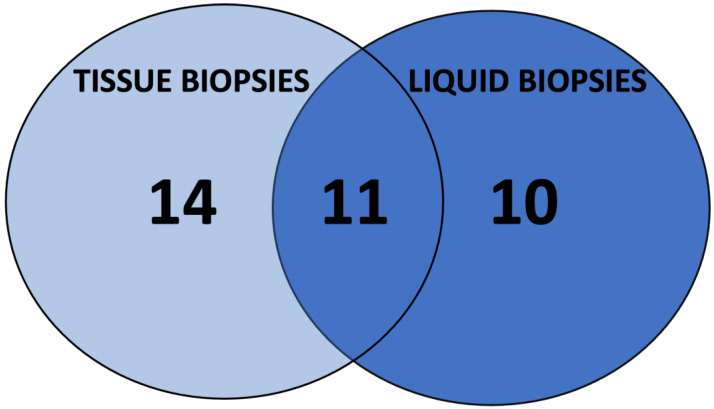
Venn Diagram showing molecular alterations found on LB and tissue biopsy.

**Table 1 cancers-14-02774-t001:** Main clinical characteristics of the patients.

**Characteristics**	**Values**	
Age (Years, Median, Range)	11 (1.5–17)	
Gender (M/F)	27/18	
First line/relapse	2/45	
Presence of metastasis at the time of LB	43	
**According to Pathology**	**Number of Patients**	**Successful LB n (%)**
Neuroblastoma	12	10 (83%)
Bone tumors	12	8 (66%)
Ewing’s sarcoma	5	4 (80%)
Osteosarcoma	7	4 (57%)
Soft Tissue Sarcomas	9	3 (33%)
Rhabdomyosarcoma	3	2
DSRCT	2	1
Malignant rhabdoid tumor	1	-
MPNST	1	-
BCOR fibromyxoid sarcoma	1	-
Intra-cardiac sarcoma	1	-
CNS tumors	7	1 (14%)
Medulloblastoma	4	1
Unclassified embryonal tumor	1	-
High-grade glioma	2	-
Other tumors	5	3 (60%)
Melanoma	2	1
UCNT	1	1
Adenocarcinoma	1	1
Teratoma	1	-

Abbreviations used: DSCRT, desmoplastic small round cell tumor; MPNST, malignant peripheral nerve sheath tumor; UCNT, undifferentiated carcinoma of the nasopharynx; LB, liquid biopsy.

## Data Availability

The data will be available upon reasonable request to the authors.

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
