# Peer review of "Blood-Derived Liquid Biopsies Using Foundation One® Liquid CDx for Children and Adolescents with High-Risk Malignancies: A Monocentric Experience"

_cancers, 2022, doi:10.3390/cancers14112774_

Round 1

Reviewer 1 Report

In this manuscript, the authors describe a retrospective cohort of 45 pediatric solid tumor patients who received a liquid biopsy test as performed by Foundation medicine. The authors report that in their diverse group by tumor histology, they could run the test on 73% of patients and that mutations/alterations were found in 55% and 38% of patients had findings that were targetable. Six patients received new therapy based on these results. The results of this study are consistent with what would be expected in an analyses of these diseases using this assay and while they are believable, do not seem to break much new ground. Additional specific concerns are as follows:

1) Please include a statement that this study was approved by a local ethics boards.

2) The authors note that they used the OncoKB database to determine if a finding was tractable. However, no data are provided as to how this resource was used. Did the mutations found in the report have to match those are considered oncogenic in the database or have been used as inclusion/exclusion criteria for previous trials for which these drugs have been studied and evidence levels collected. Did the authors consider the level of evidence in the database when deciding whether a finding was indeed targetable and that there was compelling evidence to target that finding? I suspect many of these matches were tier 4.

3) Please more carefully describe the mutations that were found and consider an oncoprint type plot which would be considered common for this kind of study.

4) Please include more details for the six patients treated with "matched therapy." What was the specific mutation and why was that drug combination chosen? What was the rationale for immunotherapy for the patient with melanoma other than published efficacy data? Why was a PARP inhibitor given to the patient with Ewing sarcoma without a clear mutation in a HR DDR pathway? A swimmers plot would be also helpful.

5) Can the authors clarify whether a molecular tumor board discussed these findings before initiating therapy?

6) The use of the Foundation liquid biopsy test seems ill-suited to pediatric cancer. Many of the common genes evaluated are not on the panel and other important drivers such as copy number and fusions either for detection and tracking as well as therapeutic decision making as the fusion Tyrosine-Kinases are the most effectively treated.

7) How did the authors choose TMB >10? This number has not really been validated in pediatric tumor biopsies and is certainly not used for liquid biopsies. What was the actual bTMB as reported by this test for these patients?

8) More consideration should be given to the utility, or more likely lack thereof, of targeting specific mutations in these pathways in pediatric solid tumors. For example, the pediatric match study recently published that single agent MEK inhibition was not effective for RAS/MAPK mutated tumors (PMID: 35363510).

Author Response

Reviewer 1

  • Please include a statement that this study was approved by a local ethics boards.

 This statement is already included in the manuscript in the dedicated section at the end of the manuscript as per instructions to authors. We obtained the approval of our institution to perform this retrospective analysis (AP-HM 2021-73).

 All parents/patients signed consent for the molecular testing though the FMI test. we obtained the approval of our institution to perform this retrospective analysis (AP-HM 2021-73).

  • The authors note that they used the OncoKB database to determine if a finding was tractable. However, no data are provided as to how this resource was used. Did the mutations found in the report have to match those are considered oncogenic in the database or have been used as inclusion/exclusion criteria for previous trials for which these drugs have been studied and evidence levels collected. Did the authors consider the level of evidence in the database when deciding whether a finding was indeed targetable and that there was compelling evidence to target that finding? I suspect many of these matches were tier 4.

  This comment raises the debate on the definition of “actionable” or “potentially actionable” alterations, or “ready for use” alterations.  These terms have been used inconsistently in the pediatric the programs. To be consistent with the French national strategy We used the same strategy proposed by E-Smart and reported by Berlanga and al. 2022.     In this report, 96% of the reported “potentially actionable” oncogenic events matched with treatment suggestions were considered at an “investigational” (80%) or “hypothetical” (16%) evidence level.  Of note, while these alterations are well-known oncogenic events, their direct targeting, alone or in combination has not been demonstrated with significant clinical activity yet. Thus, this section has been slightly rephrased.

  • Please more carefully describe the mutations that were found and consider an oncoprint type plot which would be considered common for this kind of study.

 We do agree with the reviewer. An oncoprint type figure has been added. To also answer to the comment of reviewer 2, molecular alterations found at the tumor level when paired-tumor tissue profiling was also performed have also been added.

  • Please include more details for the six patients treated with "matched therapy." What was the specific mutation and why was that drug combination chosen? What was the rationale for immunotherapy for the patient with melanoma other than published efficacy data? Why was a PARP inhibitor given to the patient with Ewing sarcoma without a clear mutation in a HR DDR pathway? A swimmers’ plot would be also helpful.

 As some of the patients were enrolled in a still ongoing or not yet published clinical trials, details about treatment cannot be provided here through a swimmer plot. We added some more detail to justify the use of immunotherapy in the patient with melanoma and explained the use of PARP inhibitor for Ewing sarcoma patient. The presence of the EWS-FLI1 fusion was a criterion of molecular enrichment for the irinotecan-olaparib arm of the E-Smart trial.

  • Can the authors clarify whether a molecular tumor board discussed these findings before initiating therapy?

 Yes, a molecular board discussed the cases. Depending on the case this was a international board  (E-SMART trial) or national board (France Medecine Genomique) or institutional board. This has been added in the text. Of note, the report generated by FMI also indicate the potential trial corresponding to each “targetable” alterations found.

  • The use of the Foundation liquid biopsy test seems ill-suited to pediatric cancer. Many of the common genes evaluated are not on the panel and other important drivers such as copy number and fusions either for detection and tracking as well as therapeutic decision making as the fusion Tyrosine-Kinases are the most effectively treated.

 We do agree with this comment: FMI liquid biopsy test analyses mutations, substitutions, copy number, rearrangements and fusions and covers over 340 genes. although EWS-based fusion for Ewing or Small Cell Desmoplastic Tumors or Myc amplification for neuroblastoma were for instance identified in our patients. We do agree anyhow that copy number and some fusions can be missed. We therefore added a section on this issue in the discussion.

  • How did the authors choose TMB >10? This number has not really been validated in pediatric tumor biopsies and is certainly not used for liquid biopsies. What was the actual bTMB as reported by this test for these patients?

 We used the TMB > 10 cut off as an extrapolation from adults’ data. Although not validated in children this is the commonly cut off used in pediatric reports (Berlanga Cancer Discovery 2022, Gubner Nature 2018 ) while others indeed chose 5 (Khater F, Vairy S, Langlois S, et al. Molecular Profiling of Hard-to-Treat Childhood and Adolescent Cancers. JAMA Netw Open. 2019;2(4):e192906. Although, we do agree with the reviewer, the is no formal comparison of TMB evaluated though tissue versus liquid biopsy in children. We have added in the oncoplot the details of TMB evaluated though LB. Unfortunately, only 4 patients had also an evaluation of the TMB at the tissue levels. We decided not to include the details and compare the results because of the very low number of cases and as all the value we low not providing possibilities to discriminate differences between LB and TB. This has been updated in the dedicated section.

  • More consideration should be given to the utility, or more likely lack thereof, of targeting specific mutations in these pathways in pediatric solid tumors. For example, the pediatric match study recently published that single agent MEK inhibition was not effective for RAS/MAPK mutated tumors (PMID:35363510).

 we do agree that identifying specific mutations or pathways that can indeed lead to efficient/clinical meaningful targeting is a crucial issue and still work in progress. We added a section about this issue in the discussion and added the suggested reference.

Reviewer 2 Report

This study reports clinically highly relevant data obtained from ctDNA diagnostics employing peripheral blood from children and adolescents with cancers.

My comments are:

1) Please improve English language and style of Figure 1.

2) Please update the design of Table 1.

3) Please report matching tumor/liquid biopsy data in Table 1. If I understand the data presented correctly, the molecular tumor profiling data are not depicted. To judge the benefit of liquid biopsy-based diagnostics, a matched data analysis (tumor/liquid biopsy) is necessary.

4) The same is true for Figure 2. This meta-analysis could be considerably improved by including the molecular tumor data.

Author Response

1) Please improve English language and style of Figure 1.

 We do agree- Figure 1 has been re-designed and English edited.

2) & 3) Please update the design of Table 1. Please report matching tumor/liquid biopsy data in Table 1. If I understand the data presented correctly, the molecular tumor profiling data are not depicted. To judge the benefit of liquid biopsy-based diagnostics, a matched data analysis (tumor/liquid biopsy) is necessary.

 In line with the comment 3 of reviewer 1, we have turned this table into an Oncoprint. Moreover to respond to comment. 3 of reviewer 1, we have collected the data from profiling of Tissue Biopsies form patients who underwent both LB and TB. These data are also included in the oncoprint. To make it easier to compare the overall results using a Venn diagram of molecular alteration detected in each type of paired samples.

4) The same is true for Figure 2. This meta-analysis could be considerably improved by including the molecular tumor data.

 The details of the molecular alterations have been added in the oncoprint.  

Round 2

Reviewer 1 Report

The authors have adequately addressed all concerns.